



# BESS-STAIR: a framework to estimate daily, 30-meter, and all-weather crop evapotranspiration using multi-source satellite data for the U.S. Corn Belt

Chongya Jiang[1,2], Kaiyu Guan[1,2,3], Ming Pan[4], Youngryel Ryu[5], Bin Peng[1,3], Sibo Wang[3]

[1]College of Agricultural, Consumer and Environmental Sciences, University of Illinois at Urbana Champaign, Urbana, Illinois, USA
[2]Center for Advanced Bioenergy and Bioproducts Innovation, University of Illinois at Urbana Champaign, Urbana, Illinois, USA
[3]National Center of Supercomputing Applications, University of Illinois at Urbana Champaign, Urbana, Illinois, USA
[4]Department of Civil and Environmental Engineering, Princeton University, New Jersey, USA
[5]Department of Landscape Architecture and Rural Systems Engineering, Seoul National University, Seoul, Republic of Korea

*Correspondence to*: Chongya Jiang (chongya.jiang@email.com)

**Abstract.** With increasing crop water demands and drought threats, mapping and monitoring of cropland evapotranspiration (ET) at high spatial and temporal resolutions becomes increasingly critical for water management and sustainability. However, estimating ET from satellite for precise water resources management is still challenging due to the limitations in both existing ET models and satellite input data. Specifically, the process of ET is complex and difficult to model, and existing satellite remote sensing data could not fulfill high resolutions in both space and time. To address the above two issues, this study presented a new high spatiotemporal resolution ET mapping framework, i.e., BESS-STAIR, which integrates a satellite-driven water-carbon-energy coupled biophysical model BESS (Breathing Earth System Simulator) with a generic and fully-automated fusion algorithm STAIR (SaTellite dAta IntegRation). In this framework, STAIR provides daily 30-meter multispectral surface reflectance by fusing Landsat and MODIS satellite data to derive fine-resolution leaf area index and visible/near-infrared albedo, all of which, along with coarse-resolution meteorological and $CO_2$ data, are used to drive BESS to estimate gap-free 30-m resolution daily ET. We applied BESS-STAIR from 2000 through 2017 in six areas across the U.S. Corn Belt, and validated BESS-STAIR ET estimations using flux tower measurements over 12 sites (85 site-years). Results showed that BESS-STAIR daily ET achieved an overall $R^2 = 0.75$, with RMSE = 0.93 mm d$^{-1}$ and relative error = 27.9% when benchmarked with the flux measurements. In addition, BESS-STAIR ET estimations well captured the spatial patterns, seasonal cycles, and interannual dynamics in different sub-regions. The high performance of the BESS-STAIR framework is primarily resulted from: (1) the implementation of coupled constraints on water, carbon, and energy in BESS, (2) high-quality daily 30-m data from STAIR fusion algorithm, and (3) BESS's applicability under all-sky conditions. BESS-STAIR is calibration-free and has great potentials to be a reliable tool for water resources management and precision agriculture applications for the U.S. Corn Belt, and even for worldwide given the global coverage of its input data.



## 1 Introduction

Accurate field-level management of water resources urgently demands reliable estimations of evapotranspiration (ET) at high spatial and temporal resolutions. ET is the sum of water loss from soil surface through evaporation and that from plant components through transpiration, and ET at cropland is usually considered as crop water needs (Allen et al., 1998). ET consumes up to 90% of total water inputs (precipitation plus irrigation) in agro-ecosystems in the Western and Midwestern United States (Irmak et al., 2012). In the U.S. Corn Belt, increasing vapor pressure deficit (VPD) and drought sensitivity has

been recognized as severe threats to future crop security (Lobell et al., 2014; Ort and Long, 2014). The vulnerability to drought in this region is further exacerbated by elevated rates of grass-to-crop conversion and expansion of irrigated areas (Brown and Pervez, 2014; Wright and Wimberly, 2013). Furthermore, precision water resources management requires capacity to account for spatial heterogeneity and to guide real-time decision-making (USDA, 1997). Accordingly, reliable tools are urgently needed to estimate, map and monitor the total amount and spatial and temporal variations of cropland ET.


One critical requirement for the accurate estimations of ET at high spatiotemporal resolutions is reliable and advanced satellite-based models. This is challenging because the process of ET is complex and difficult to model. ET results from competitions between atmospheric water demand and soil water supply, and it is also regulated by plants through canopy development and stomatal behaviors in order to optimize their water, carbon and energy use strategies (Katul et al., 2012;

Wang and Dickinson, 2012). A large number of satellite-based ET estimation methods have been developed based on different theories and techniques. In general, they can be grouped into many categories: statistical or machine learning methods (Jung et al., 2010; Lu and Zhuang, 2010), water balance methods (Pan et al., 2012; Wan et al., 2015), energy balance methods (Anderson et al., 1997; Su, 2002), triangular or trapezoid space methods (Jiang and Islam, 1999; Li et al., 2009), Priestley–Taylor methods (Fisher et al., 2008; Miralles et al., 2011), and Penman–Monteith methods (Mu et al., 2011;

Yebra et al., 2013). Kalma et al. (2008), Li et al. (2009) and Zhang et al. (2016) have provided detailed reviews on the pros and cons of different remote sensing approaches.

Given the complexity of the ET process, we argue that a reliable ET model should include both necessary biophysical processes and high-quality multi-source observations to constrain ET estimations (Loew et al., 2016). While remote sensing-

based approaches tend to focus on constraints from various satellite data, land surface models (LSMs) are proficient to include processes that account for interactions between environment and plant structure and functions. Given the gaps between remote sensing and LSMs, a distinct ET model, the Breathing Earth System Simulator (BESS), was developed (Jiang and Ryu, 2016; Ryu et al., 2011). Different from the above-mentioned remote sensing models, BESS is a biophysical model, which adopts modules commonly-implemented in land surface models but uses various satellite remote sensing data

as direct inputs. Specifically, BESS is a two-leaf water-carbon-energy coupled model driven by environmental and vegetation variables derived from multi-source satellite data. As the energy cycle, carbon cycle and water cycle are jointly modeled and mutually constrained in BESS, it has produced a series of high-quality global long-term (2000-2017) products,



including the 5-km resolution global radiation (Rg) and photosynthetically active radiation (PAR) and diffuse PAR products (Ryu et al., 2018), and 1-km resolution gross primary productivity (GPP) and ET products (Jiang and Ryu, 2016), which

enables tracking crop growth and yields too (Huang et al., 2018). In particular, the 1-km resolution BESS ET product is able to capture the total amount and spatial and temporal variations in arid/semi-arid areas like Australia (Whitley et al., 2016, 2017), California (Baldocchi et al., 2019) and Northwestern China (Wei et al., 2019). The fidelity of coarse-resolution BESS ET product suggests its potential at fine resolutions.

The other critical requirement for accurate estimations of ET at high spatiotemporal resolutions is satellite input data at high resolutions in both space and time. This is challenging because existing satellite missions cannot satisfy the two conditions simultaneously. Data fusion techniques, which take multi-sensor data to generate fusion data with high resolutions in both space and time, provide a possible and scalable solution. Several such algorithms have been developed over the past decade (Gao et al., 2006; Houborg and Mccabe, 2018; Zhu et al., 2010), and they have been successful for localized applications

(Gao et al., 2017; Gómez et al., 2016; Wu et al., 2015). Notably, energy balance and thermal-based ET models such as ALEXI/DisALEXI and SEBS have been combined with the fusion algorithm such as STARFM and ESTARFM to generate daily 30-m ET estimations with favourable performance at several sites (Anderson et al., 2018; Cammalleri et al., 2013; Li et al., 2017; Ma et al., 2018).

Here we propose and present a new ET estimation framework that combines BESS with a novel fusion algorithm SaTallite dAta IntegRation (STAIR) (Luo et al., 2018), for accurate ET estimation at both high resolution in time and space. BESS has demonstrated its high performance in estimating ET at medium to coarse resolutions, but the major obstacle of moving BESS's ET estimation to finer resolutions is the lack of key vegetation status variables at higher spatial resolutions, including leaf area index (LAI), and visible and near-infrared albedo ($\alpha_{VIS}$ and $\alpha_{NIR}$). In BESS, these surface information are

critical to resolving spatial heterogeneity, while environmental information such as radiation, temperature, humidity and $CO_2$ concentration are relatively homogeneous. To cope with the absence of high spatiotemporal resolution vegetation data, we propose to couple STAIR with BESS. STAIR is a genetic and fully-automated fusion algorithm to generate cloud-/gap-free surface reflectance product in high spatiotemporal resolution (Luo et al., 2018). Instead of manually selecting image pairs adopted by most other data fusion algorithms, STAIR automatically takes full advantage of time-series of daily coarse-

resolution images and fine-resolution but less frequent images. Moreover, STAIR's high efficiency in computation allows scalability for large scale productions, which enable this new framework to deliver daily 30-m ET at regional and decadal scales.

The objective of this study is to address a fundamental issue in agro-ecological science and applications: lack of high

spatiotemporal gap-free ET data for decision-making. We implemented a new ET estimation framework BESS-STAIR and tested it at six study areas across the U.S. Corn Belt from 2000 to 2017. This is the first attempt to couple a satellite-driven



LSM with data fusion technique to provide daily 30m-resolution ET estimations at regional and decadal scales. While existing frameworks retrieve clear-sky ET from satellite-observed LST and fill ET gaps for cloudy-sky days, BESS-STAIR simulates all-sky ET and LST as a result of crop biophysical properties. This manner has more referential significance for

crop modeling studies and has potential of breaking a new path to agro-ecological science and applications. We conducted comprehensive evaluation on the BESS-STAIR ET estimations with regards to the overall performance, spatial patterns, seasonal cycles and interannual dynamics, benchmarked on the ET observations from 12 eddy-covariance flux towers across the U.S. Corn Belt. The paper also discussed on the performance, advantages, limitations and potential improvements of the BESS-STAIR ET framework.


## 2 Materials and methods

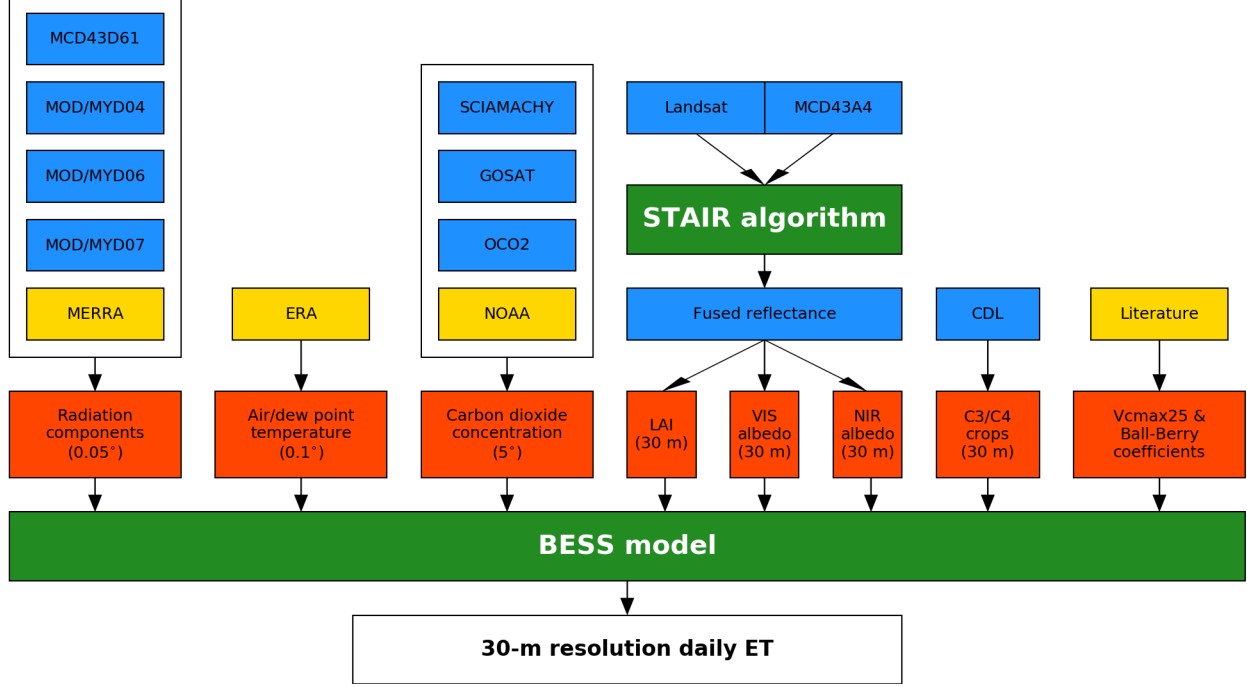

Figure 1. The BESS-STAIR framework. The BESS ET estimation model and the STAIR data fusion algorithm are highlighted in green boxes. Blue boxes are satellite data, yellow boxes are ancillary data, and red boxes are key inputs to

BESS. The output of BESS-STAIR is the 30-m resolution daily ET highlighted in white box.

BESS-STAIR estimates cropland ET at 30-m resolution at daily interval (Figure 1). BESS is driven by environmental variables (radiation, temperature, humidity, and $CO_2$ concentration), plant structural variables (LAI, $\alpha_{VIS}$ and $\alpha_{NIR}$), and plant functional variables (peak maximum carboxylation rate at 25 °C (peak $V_{cmax25}$) and Ball-Barry coefficients, for C3 and C4

plants, respectively). Among these key inputs, LAI, $\alpha_{VIS}$ and $\alpha_{NIR}$ characterize crop canopy structure, which are usually very





heterogeneous. In the global BESS ET product (Jiang and Ryu, 2016), these vegetation variables are derived from MODIS satellite data at 1-km resolution; while in BESS-STAIR, they are derived from 30-m resolution surface reflectance fused from high spatial resolution Landsat data and high temporal resolution MODIS data by STAIR.

## 2.1 The ET estimation model: BESS

BESS is a sophisticated satellite-driven water-carbon-energy coupled biophysical model designed to continuously monitor and map water and carbon fluxes (Jiang and Ryu, 2016; Ryu et al., 2011). It is a simplified land surface model, including an atmosphere radiative transfer module (Kobayashi and Iwabuchi, 2008; Ryu et al., 2018), a two-leaf canopy radiative transfer module (De Pury and Farquhar, 1997), and an integrated carbon assimilation – stomatal conductance – energy balance module. Specifically, the Farquhar model for C3 and C4 plants (Collatz et al., 1992, 1991), the Ball-Berry model (Ball et al., 1987), and the quadratic form of the Penman-Monteith equation (Paw U and Gao, 1988) are used for the simulation of carbon assimilation, stomatal conductance and energy balance, respectively. This carbon-water integrated module employs an iterative procedure to solve intercellular $CO_2$ concentration, stomatal conductance and leaf temperature for sunlit and shade canopy. Instantaneous sunlit/shade GPP and sunlit/shade/soil ET and net radiation at Terra and Aqua overpass times are simultaneously estimated, followed by a temporal upscaling procedure to derive daily GPP and ET using semi-empirical cosine functions (Ryu et al., 2012). The Priestley-Taylor equation is used to compute daily potential ET (PET) based on daily net radiation and meteorological data.

A unique feature of BESS is that BESS takes full advantages of atmospheric and land products derived from multi-source satellite data. By using MOD/MYD 04 aerosol products (Sayer et al., 2014), MOD/MYD 06 cloud products (Baum et al., 2012), MOD/MYD 07 atmospheric profile products (Seemann et al., 2003), along with gap-free atmospheric data provided by MERRA-2 reanalysis products (Gelaro et al., 2017), BESS calculates direct/diffuse visible/near-infrared radiation components at 0.05° resolution. By coupling $CO_2$ concentration derived from SCIAMACHY and GOSAT satellite data (Dils et al., 2014) with those from OCO-2 satellite data (Hammerling et al., 2012), as well as NOAA long-term field observations (www.esrl.noaa.gov/gmd/ccgg/trends/), BESS derives long-term continuous monthly $CO_2$ concentration maps. Finally, in this study BESS uses air temperature and dew point temperature provided by ERA5 reanalysis products at 0.1° resolution (Hersbach and H., 2016). In addition to these environmental variables, BESS also highly relies on vegetation structural and functional variables. By using satellite-derived LAI, $\alpha_{VIS}$ and $\alpha_{NIR}$, BESS quantifies the absorption of ultraviolet/visible/near infrared radiation by sunlit/shaded canopy through a canopy radiative transfer model. This model also upscales leaf level ($V_{cmax25}$) to sunlit/shade canopy, which is used in the Farquhar photosynthesis model. $V_{cmax25}$ is a parameter depending on the plant functional type (Bonan et al., 2011; Kattge et al., 2009), and its seasonal variation is empirically parameterized by LAI (Ryu et al., 2011).





## 2.2 The data fusion algorithm: STAIR

STAIR is a generic and fully-automated method for fusing multi-spectral satellite data to generate high spatiotemporal resolution and cloud-/gap-free data (Luo et al., 2018). It fully leverages the complementary strengths in the high temporal resolution MCD43A4 nadir reflectance (daily but 500 m resolution) (Schaaf et al., 2002) and the high spatial resolution Landsat L2 nadir reflectance (30-m resolution but 16-day revisiting frequency) (Masek et al., 2006) time series data. STAIR first imputes the missing pixels using an adaptive-average correction procedure, and then employs a local interpolation

model to capture finer spatial information provided by Landsat data, followed by a time-series refinement step that incorporates the temporal patterns provided by MODIS data. This strategy allows higher efficiency in missing-data interpolation as well as greater robustness against concurrently missed MODIS and Landsat observation, which is a common situation during continuous cloudy/snowy days.

The algorithm starts from the imputation of the missing pixels (due to cloud cover or Landsat 7 Scan Line Corrector failure) in satellite images. For MODIS images, a Savitzky-Golay filter is first applied to reconstruct continuous time series. For Landsat images, a two-step approach is employed using both temporal and spatial information from clear-sky observations. First, a temporal interpolation through a linear regression is applied as the initial gap-filling, based on the whole time series of images throughout a year. Second, an adaptive-average correction procedure is applied to remove inharmonic spatial

patterns between gap-filled and original data. The target image is partitioned into multiple segments, each of which contains one type of homogeneous pixels. The relative difference between a gap pixel and neighbourhood pixels of it within the same segment is calculated using clear-sky observations acquired in several dates close to the target image acquisition date. Based on the assumption that the relative difference remains roughly the same across different dates in a short time period (e.g., < 2–3 weeks), such difference is used to correct the filled values of the gap pixel derived from temporal interpolation so that

the spatial relationship between the gap-filled pixel and its neighbourhood pixels within the same segment is consistent with those in clear-sky observations.

The STAIR fusion algorithm fully exploits the spatial and temporal information in the time series of gap-filled MODIS and Landsat images throughout the growing season (April - October). A nearest neighbour sampling is conducted for all the

MODIS images to achieve the same image size, pixel resolution and projected coordinate system with Landsat images. Difference image is calculated for each pair of Landsat and resampled MODIS images, and a linear interpolation is applied to reconstruct the difference image for any given date when no Landsat image is available. Such difference image is used to correct the resampled MODIS image on that date and to generate a fused Landsat image. In this manner, the fused image captures the most informative spatial information provided by the high spatial resolution Landsat data and incorporates the

temporal patterns provided by the high temporal resolution MODIS data without any user interference. The fusion algorithm is applied to the six Landsat bands: blue, green, red, near-infrared (nir), shortwave infrared 1 (swir1), and shortwave infrared-2 (swir2).





### 2.3 Derivation of BESS inputs from STAIR data

At global scale, LAI, $\alpha_{VIS}$ and $\alpha_{NIR}$ can be obtained from MODIS and other satellite data, but for field-scale agricultural
applications high spatial resolution data are needed to account for the spatial heterogeneity between fields or within a field.
At this point, we employed two approaches to estimate 30-m resolution daily LAI from STAIR fused surface reflectance
data: an empirical approach based on linear relationship with vegetation indices (VIs) and a mechanic approach based on
inversion of a canopy radiative transfer model (RTM).


First, we estimated LAI using the empirical approach, because of availability of field LAI measurements in the study area.
We calculated four VIs calculated from STAIR-derived spectral reflectance: Wide Dynamic Range Vegetation Index
(WDRVI), Green Wide Dynamic Range Vegetation Index (GWDRVI), Enhanced Vegetation Index (EVI), and Land Surface
Water Index (LSWI) for corn and soybean, respectively (Eq. (1) – (3)). These four Vis were chosen because they utilized
information from different band combinations.

$$WDRVI = \frac{0.1\rho_N - \rho_R}{0.1\rho_N + \rho_R} \tag{1}$$

$$GWDRVI = \frac{0.1\rho_N - \rho_G}{0.1\rho_N + \rho_G} \tag{2}$$

$$EVI = 2.5 \frac{\rho_N - \rho_R}{\rho_N + 6\rho_R - 7.5\rho_B + 1} \tag{3}$$

$$LSWI = \frac{\rho_N - \rho_{SW1}}{\rho_N + \rho_{SW2}} \tag{4}$$

where $\rho_B$, $\rho_G$, $\rho_R$, $\rho_N$, and $\rho_{SW1}$ refer to the surface reflectance at blue, green, red, near-infrared, the first shortwave-infrared
band, respectively. Subsequently, we used field measured LAI data collected using destructive method at Mead, Nebraska
from 2001 through 2007 to build VI-LAI relationships (Gitelson et al., 2007). For each of the four VIs we build a linear
regression between time series of VI and LAI for corn, soybean, and the combination of corn and soybean, respectively. At
this point, the equation derived from the combination of corn and soybean was used for vegetation cover other than corn and
soybean. Although this might cause bias for forest LAI estimation, it is not a concern in this study as we focused on crop ET
only. We applied linear regressions to four VIs separately and averaged the four derived LAI as the final LAI estimation,
with the expectation that such average would reduce uncertainty caused by individual VI-LAI relationship.

Second, we inversed PROSAIL RTM using a look-up table (LUT) method. PROSAIL is an efficient and widely-used model
to simulate canopy reflectance given a set of sun-object-view geometry, canopy structure, leaf biochemical, and soil optical
parameters (Jacquemoud et al., 2009). It is a combination of the PROSPECT leaf hyperspectral properties model
(Jacquemoud et al., 1996; Jacquemoud and Baret, 1990) and the SAIL canopy bidirectional reflectance model (Verhoef,


1984, 1985). PROSAIL is particularly suitable for grasslands and croplands (Darvishzadeh et al., 2008; Xu et al., 2019), and
therefore used in this study. LUT is a robust and easy method to retrieve model parameters from observed canopy reflectance
(Verrelst et al., 2018). It is based on the generation of simulated canopy reflectance database for a number of plausible
combinations of model parameter value ranges, and the identification of parameter values in the database leading to the best
agreement between simulated and observed canopy reflectance. LUT is particularly suitable for big data processing (Myneni
et al., 2002), and therefore used in this study.


We established a database by running PROSAIL with sampled parameter values listed in Table 1. For computation
efficiency, we only sampled varied values for four parameters while others were fixed. These four free parameters, including
LAI (10 values), fraction of vegetation cover (6 values), soil brightness (5 values) and chlorophyll content (4 values), were
chosen because they have been identified as the most sensitive parameters in canopy radiative transfer models (Bacour et al.,
2002; Mousivand et al., 2014). Leaf inclination distribution function is also sensitive but we set fixed types "spherical",
"planophile" and "plagiophile" for corn, soybean, and other biomes, respectively (Nguy-Robertson et al., 2012; Pisek et al.,
2013). The fixed values of other parameters were set according to literature (Baret et al., 2007; Feret et al., 2008;
Jacquemoud et al., 2009). Solar zenith angle at satellite overpass time can be calculated so we did not set it as a free
parameter. Instead, we built a set of databases with solar zenith angle values (°) of 20, 25, 30, 35, 40, 45 and 50, respectively,
representing the range during growing season in the study area. In PROSAIL, specific absorption coefficients and refractive
index of leaf material are pre-measured hyperspectral data from 400 to 2500 nm with 1 nm interval (Feret et al., 2008), we
averaged them over wavelengths to match Landsat 7 bands and assumed differences of spectral ranges between Landsat 5,
Landsat 7 and Landsat 8 have marginal influence on LAI retrieval. We did not use default soil spectrum in PROSAIL, but
spatiotemporally averaged all cropland pixels spectral reflectance in April when no crop is planted across the study area to
derive representative soil spectral reflectance.

Table 1. Parameter values needed to establish the canopy reflectance database by PROSAIL.

| Parameters | Values |
|---|---|
| LAI (m2 m-2) | 0, 0.5, 1, 2, 3, 4, 5, 6, 7, 8 |
| Fraction of vegetation cover (m2 m-2) | 0, 0.2, 0.4, 0.6, 0.8, 1 |
| Soil brightness (a.u.) | 0.01, 0.4, 0.8, 1.2, 1.6 |
| Chlorophyll content (ug cm-2) | 0, 20, 40, 60 |
| Leaf inclination distribution function | spherical for corn, planophile for soybean, plagiophile for others |
| Structure coefficient (a.u.) | 1.75 |
| Carotenoid content (ug cm-2) | 0 |
| Equivalent Water Thickness (cm) | 0.015 |
| Leaf Mass per area (g cm-2) | 0.0075 |



| | |
|---|---|
| Brown pigment content (a.u.) | 0 |
| Hot spot parameter (a.u.) | 0.1 |
| View zenith angle (°) | 0 |
| Azimuth angle (°) | 0 |

To retrieve LAI, we compared STAIR-derived surface reflectance ($R_{STAIR}$) with records in the canopy reflectance database
simulated by PROSAIL ($R_{PROSAIL}$) pixel by pixel. We used root mean square error (RMSE) as the cost function which was
defined as:

$$RMSE = \sqrt{\frac{1}{l}\sum_{\lambda=1}^{l}[R_{STAIR}(\lambda) - R_{PROSAIL}(\lambda)]^2},$$ (5)

where $\lambda = 1, 2, \dots l$ indicates band number and $l = 6$ for STAIR. Ideally, the simulated reflectance in the database yielding the
smallest RMSE can be considered as the best simulation, and the corresponding LAI value can be considered as the solution
for the satellite pixel. However, in reality the solution might not be unique, because different parameter combinations could
derive similar reflectance simulations and errors in both satellite and model could further amplify this problem (Verrelst et
al., 2018). For this reason, we chose top 10% small RMSE simulations in the database and considered the average of
corresponding LAI values as the final solution. The threshold 10% was decided by evaluating LUT-retrieved LAI against
field-measured LAI at three Mead sites, and it was within a reasonable range from top 50 records to top 20% records
suggested by previous studies (Duan et al., 2014; Weiss et al., 2000).

We further employed semi-empirical equations to calculate $\alpha_{VIS}$ and $\alpha_{NIR}$ (Liang, 2001) from STAIR-derived spectral
reflectance in six Landsat bands:

$$\alpha_{VIS} = 0.443\rho_B + 0.317\rho_G + 0.240\rho_R$$ (6)

$$\alpha_{NIR} = 0.693\rho_N + 0.212\rho_{SW1} + 0.116\rho_{SW2} - 0.003$$ (7)

where $\rho_{SW2}$ is the surface reflectance at the second shortwave-infrared band. CI was set 0.75 for herbaceous and 0.70 for
woody plants according to the global mean value of different plant functional types (He et al., 2012). For C3 crops/grasses,
forests, and C4 crops/grasses, peak $V_{cmax25}$ values were set 180, 60 and 45, respectively (Kattge et al., 2009; Zhang et al.,
2014). Ball-Berry slope and intercept are another two important parameters used in the stomatal conductance model, and
their values were set 13.3 and 0.02 for C3 crops/grasses, 9.5 and 0.005 for forests, and 5.8 and 0.04 for C4 crops,
respectively (Miner et al., 2017). Distributions of C3 and C4 crops were obtained from Crop Data Layer (CDL) data (Boryan
et al., 2011).

**2.4 Evaluation of BESS-STAIR ET**

The BESS-STAIR ET estimations were evaluated against flux tower ET measurements in the U.S. Corn Belt. The U.S. Corn
Belt (Figure 2) generally refers to  a region in the Midwestern United States that has dominated corn and soybean production





in the United States (Green et al., 2018), which currently produces about 45% and 30% of the global corn and soybean,
respectively (USDA, 2014). The region is characterized by relatively flat land, deep fertile soils, and a high soil organic
matter (Green et al., 2018). Most part of the U.S. Corn Belt has favorable growing conditions of temperature and rainfall. A
majority of the croplands in the U.S. Corn Belt are rainfed, with a small portion in the west part relying on irrigation.

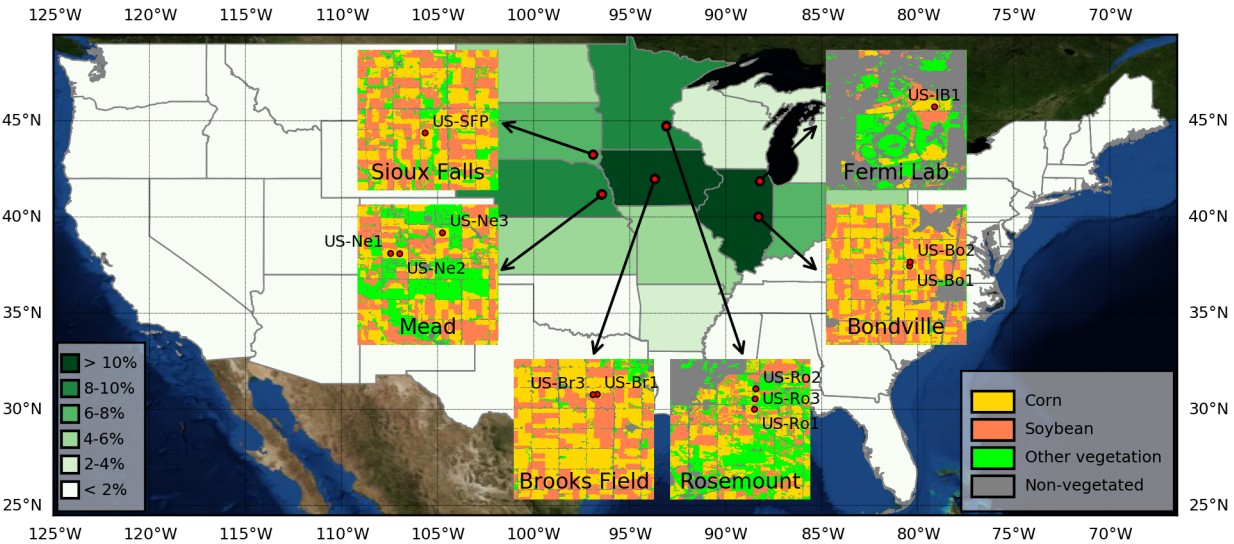

Figure 2. Study areas. Red dots indicate 12 flux tower sites scattered in six areas across the U.S. Corn Belt. The background
map indicates the percent each state contribute to the total national corn and soybean plantation area (USDA, 2018).

A total of 12 cropland sites scattered in six areas across the U.S. Corn Belt are registered in the AmeriFlux or FLUXNET
network with publicly-available ET data (Figure 2 and Table 2). These sites include both corn only and corn/soybean

rotation sites and both rainfed and irrigated sites, covering typical cropping patterns in the U.S Corn Belt. All of them were
used in this study to ensure the representativeness of the validation for the precision agriculture applications in this region.
For six sites: US-Bo1 (Meyers and Hollinger, 2004), US-Bo2 (Bernacchi et al., 2005), US-Br1 (Prueger et al., 2003), US-
Br3 (Prueger et al., 2003), US-Ro2 (Turner et al., 2016) and US-SFP (Wilson and Meyers, 2007), level 2 half-hourly data
were downloaded from the AmeriFlux website (http://ameriflux.lbl.gov/). For three sites: US-IB1 (Matamala et al., 2008),

US-Ro1 (Griffis et al., 2010) and US-Ro3 (Griffis et al., 2010), standardized gap-filled level 4 daily mean data were
downloaded from the Carbon Dioxide Information Analysis Center data archive website (https://mirrors.asun.co/climate-
mirror/cdiac.ornl.gov/pub/ameriflux/). For the other three sites (Suyker et al., 2004): US-Ne1, US-Ne2 and US-Ne3,
standardized high-quality gap-filled daily mean data were downloaded from the FLUXNET2015 website
(http://fluxnet.fluxdata.org/data/fluxnet2015-dataset/).


Table 2. Information of 12 flux tower sites used for validation.





| ID | Site | Location | Latitude | Longitude | Plant | Irrigation | Years | Source |
|----|------|----------|----------|-----------|-------|------------|-------|--------|
| 1 | US-Bo1 | Bondville, IL | 40.0062 | -88.2904 | corn/soybean | N | 2000-2008 | AmeriFlux L2 |
| 2 | US-Bo2 | Bondville, IL | 40.0090 | -88.2900 | corn/soybean | N | 2004-2006 | AmeriFlux L2 |
| 3 | US-Br1 | Brooks Field, IA | 41.9749 | -93.6906 | corn/soybean | N | 2005-2011 | AmeriFlux L2 |
| 4 | US-Br3 | Brooks Field, IA | 41.9747 | -93.6936 | corn/soybean | N | 2005-2011 | AmeriFlux L2 |
| 5 | US-IB1 | Fermilab, IL | 41.8593 | -88.2227 | corn/soybean | N | 2005-2007 | AmeriFlux L4 |
| 6 | US-Ne1 | Mead, NE | 41.1651 | -96.4766 | corn | Y | 2001-2012 | FLUXNET2015 |
| 7 | US-Ne2 | Mead, NE | 41.1649 | -96.4701 | corn/soybean | Y | 2001-2012 | FLUXNET2015 |
| 8 | US-Ne3 | Mead, NE | 41.1797 | -96.4397 | corn/soybean | N | 2001-2012 | FLUXNET2015 |
| 9 | US-Ro1 | Rosemount, MN | 44.7143 | -93.0898 | corn/soybean | N | 2004-2006 | AmeriFlux L4 |
| 10 | US-Ro2 | Rosemount, MN | 44.7288 | -93.0888 | corn/soybean/clover | N | 2008-2016 | AmeriFlux L2 |
| 11 | US-Ro3 | Rosemount, MN | 44.7217 | -93.0893 | corn/soybean | N | 2004-2006 | AmeriFlux L4 |
| 12 | US-SFP | Sioux Falls, SD | 43.2408 | -96.9020 | corn | N | 2007-2009 | AmeriFlux L2 |

By comparing with eddy covariance ET, we evaluated three ET estimations: BESS-STAIR with VIs-based LAI, BESS-

STAIR with RTM-based LAI, and BESS-STAIR with MODIS LAI. MODIS LAI refers to MCD15A3H 500m resolution 4-

day composite LAI product downloaded from https://lpdaac.usgs.gov/tools/data-pool/. Since eddy covariance technique used

by flux towers provides water flux observations in term of latent heat (LE) rather than ET, evaluations were conducted by

comparing BESS-STAIR daily LE estimates with flux tower measurements. At this point, water flux can be simply

converted from energy unit (LE, MJ m$^{-2}$ d$^{-1}$) to water unit (ET mm d$^{-1}$) by dividing latent heat of vaporization which is a

function of daily temperature (Henderson-Sellers, 1984). Flux tower measurements usually have an irregular and dynamic

footprint at scales from 100-m to 1-km (Fu et al., 2014), but for simplicity, only 30-m resolution BESS-STAIR pixels

containing the flux tower were used for the direct comparison. With regard to flux towers, measurements data were directly

used without energy closure adjustment. For AmeriFlux level 2 data, half-hourly data were averaged to daily LE only if no

gaps exist during the day to avoid sampling bias caused by missing data. For AmeriFlux level 4 data and FLUXNET2015

data, gap-filled daily LE were used directly.

## 3 Results

### 3.1 Performance of STAIR LAI

LAI is the key input of BESS. The accuracy of high-resolution LAI estimations determine the validity of high-resolution ET

estimations. We evaluated VIs-based LAI and RTM-based LAI estimations derived from 30-m resolution STAIR fused

surface reflectance data against field measurements. We also compared them with 500-m resolution MODIS LAI. Overall,

STAIR-derived LAI agree well with measured LAI, with $R^2 > 0.85$, RMSE $< 0.8$ and mean bias error (MBE) $\approx 0$ (Figure 3).

The RTM-based method which is calibration free yields same performance with VIs-based method which requires

substantial field measurements to build empirical relationships. Misclassification of CDL data between corn and soybean is





an important uncertainty source since both methods rely on crop types. During 2001 – 2007, 4 out of 21 site years (19%) over the three Mead sites were misclassified. By using the correct classification (not shown), the accuracy of LAI estimations reach $R^2 = 0.90$ and RMSE = 0.62 for VIs-based method and $R^2 = 0.89$ and RMSE = 0.68 for RTM-based method. By comparison, coarse-resolution MODIS LAI has relatively large errors, especially a negative bias ($R^2 = 0.55$, RMSE = 1.68 and MBE = -0.97).


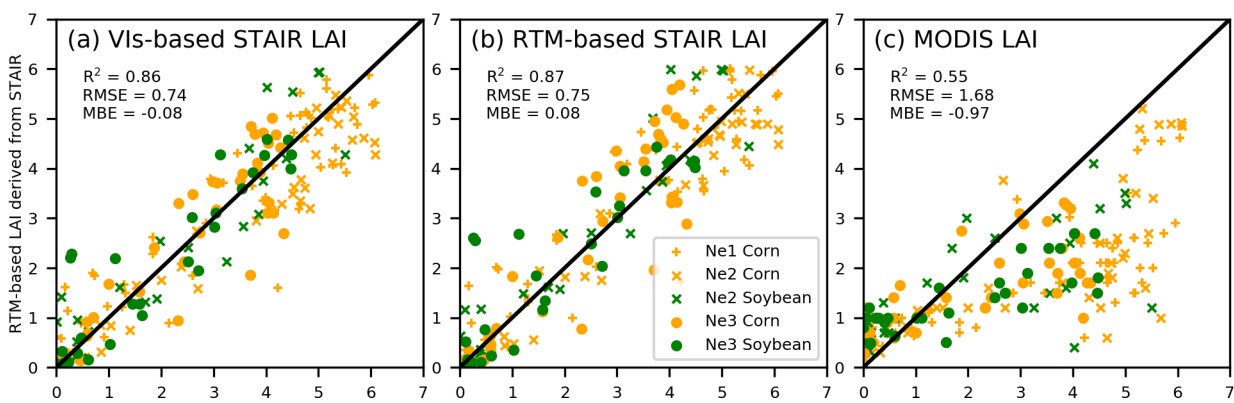

Figure 3. Scatter plots between LAI measurements and LAI estimations. LAI measurements are destructively collected at three Mead sites. (a) – (b) STAIR-derived daily 30m-resolution LAI using VIs-based method and RTM-based method, respectively. (c) 500m-resolution MODIS LAI.


### 3.2 Performance of BESS-STAIR ET

BESS-STAIR daily LE estimations are in a highly aligned agreement with ground truth from the 12 flux-tower measurements (Figure 4). Across all of the 12 sites, BESS-STAIR LE with RTM-based LAI achieves an overall coefficient of determination ($R^2$) of 0.75, root-mean-square error (RMSE) of 2.29 MJ m$^{-2}$ d$^{-1}$, relative error ($E(|X_{estimation}-$
$X_{measurement}|)/E(X_{measurement})$, RE) of 27.9%, and no overall bias. Figure 5 further exhibits its performance over all of the 12 flux tower sites. $R^2$ values range from 0.68 to 0.94 for corn, and 0.65 to 0.81 for soybean, highlighting the robustness of BESS-STAIR ET in the U.S. Corn Belt. BESS-STAIR LE with VIs-based LAI has similar performance ($R^2 = 0.75$, RMSE = 2.24 MJ m$^{-2}$ d$^{-1}$, and RE = 27.4%). Considering relatively small difference between BESS-STAIR using RTM-based LAI and that using VIs-based LAI, only the former one which is calibration-free is demonstrated in the following parts of this
paper. By comparison, BESS LE with MODIS LAI shows larger errors ($R^2 = 0.65$, RMSE = 2.50 MJ m$^{-2}$ d$^{-1}$, and RE = 30.2%) comparing to BESS-STAIR.

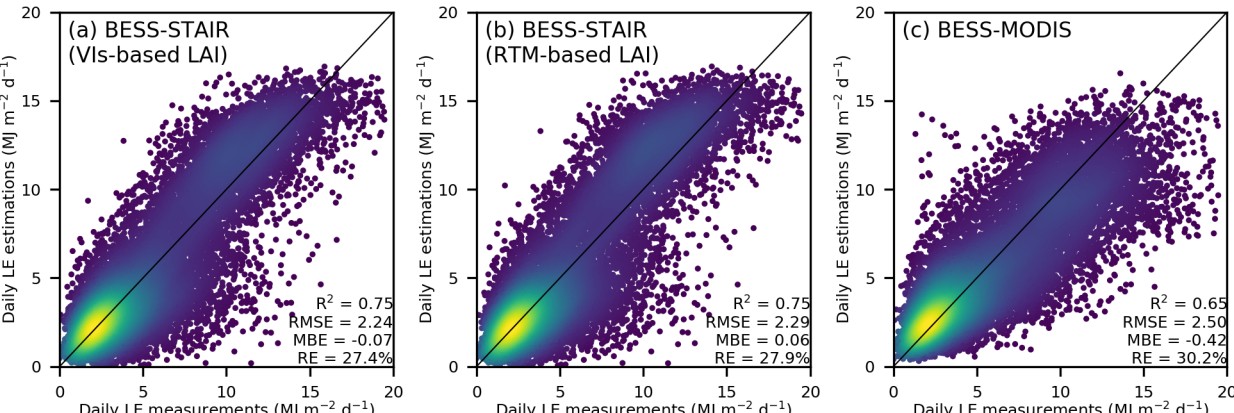

Figure 4. Density scatter plots between LE measurements and LE estimations. LE measurements are from eddy covariance

data collected at 12 flux towers. (a) and (b) BESS-STAIR LE with VIs-based LAI and RTM-based LAI, respectively. (c) 500m-BESS LE with MODIS LAI.

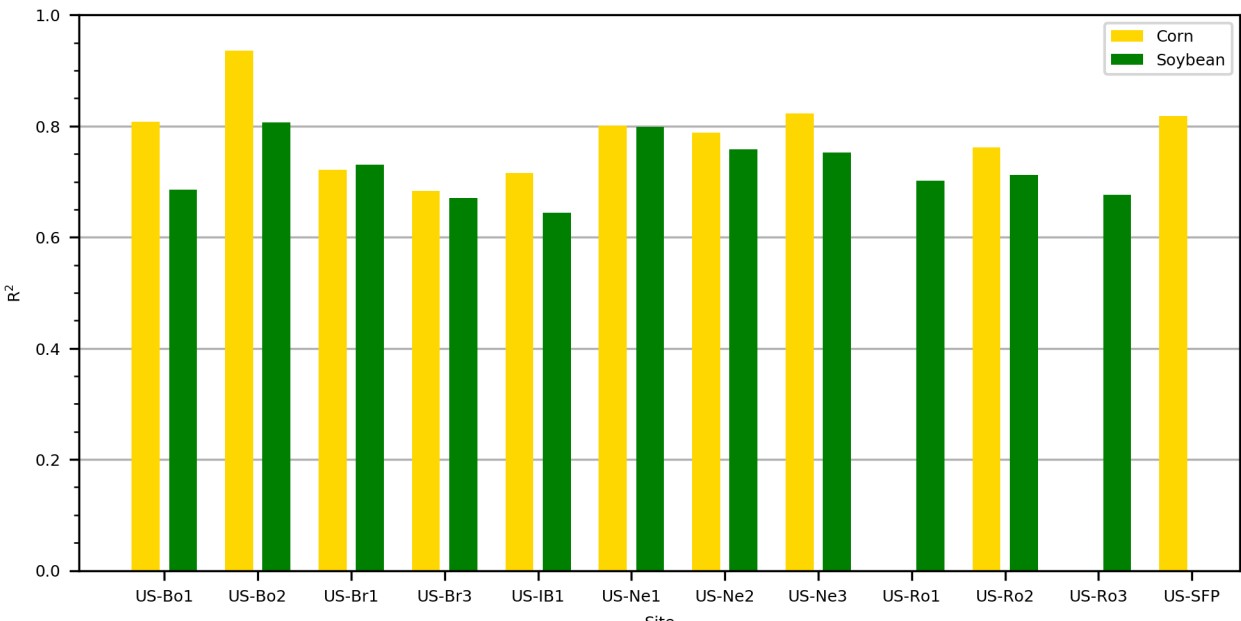

Figure 5. Site-by-site $R^2$ between flux tower measured and BESS-STAIR estimated daily LE for corn and soybean,

respectively. Crop type is from CDL data.

Figure 6 shows the comparison between BESS-STAIR daily LE estimations and flux tower measurements over site years with least data gaps in measurements. Across all of the 12 sites, BESS-STAIR well captures the seasonal characteristics of LE observation from flux towers, as they exhibit generally consistent variations over the growing season. During the peak

growing season (June, July and August), the radiation displays a dominant impact on measured daily LE, and it is reasonably





estimated by BESS-STAIR LE as well. In most cases, measured daily LE do not show strong and fast response to precipitation and/or irrigation, possibly due to the plentiful water storage in soil. Two exceptions are US-IB1 (2006) and US-Ne3 (2012). In case of US-IB1, no precipitation is available in August and little in July. As a result, daily LE measurements drop slightly quicker in August than other cases. Such anomaly is also depicted by BESS-STAIR LE. In case of US-Ne3, the

severe drought in 2012 summer causes much lower LE values than the two adjacent irrigation sites (US-Ne1 and US-Ne2). BESS-STAIR LE also captures this considerable reduction, although a slight bias is observed in July. Figure 7 further demonstrates that the seasonal cumulative ET at three Mead sites calculated for both the flux tower measurements and BESS-STAIR estimations overall agree well throughout the peak growing season (June – September).

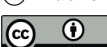





Figure 6. Seasonal time series of flux tower measured and BESS-STAIR estimated daily LE for 12 selected site years. Daily radiation and precipitation/irrigation are overlaid except for US-Br3.

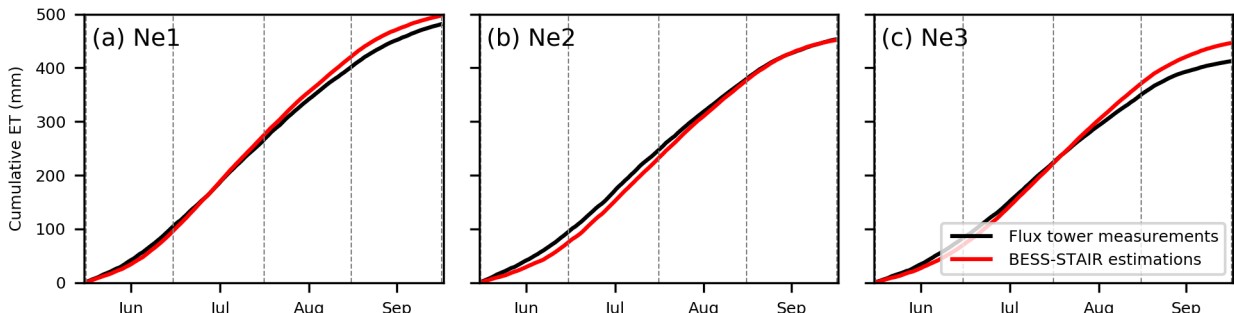

Figure 7. Multi-year mean cumulative ET for flux tower measurements and BESS-STAIR estimations at three Mead sites
from 2001 through 2012.

## 3.3 Spatiotemporal variations of BESS-STAIR ET

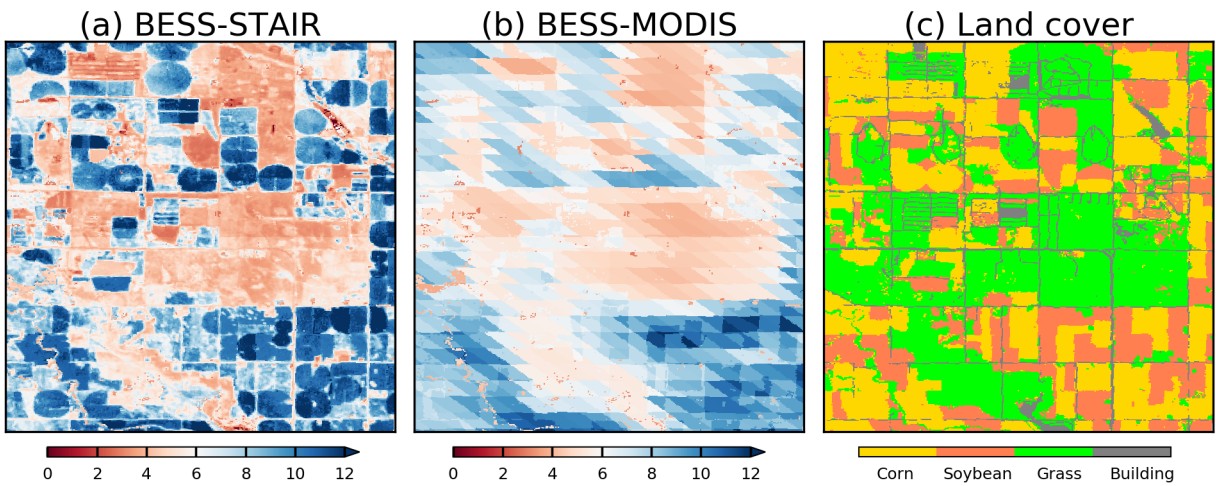

Figure 8. Daily LE (MJ m-2 d-1) derived from (a) BESS-STAIR and (b) BESS-MODIS at Mead (41.1°N – 41.2°N, 96.4°W – 96.5°W) on August 1, 2012.

BESS-STAIR daily ET demonstrates prominent spatial variations within the 0.1° × 0.1° area near the Mead site in Nebraska (Figure 8). Because of the impact of drought, central pivot irrigated fields characterized by round-shaped plots generally
display higher values than surrounding croplands, and croplands have much higher values than grasslands. Variabilities of





ET between different crop fields and within individual crop fields are also observable. Such variabilities might be attributed to different irrigation strategies, varieties and/or other management. By comparison, though 500m-resolutuion BESS-MODIS ET is able to capture the general spatial pattern, it has many mixed pixels and is unable to demonstrate gradients across field boundaries.


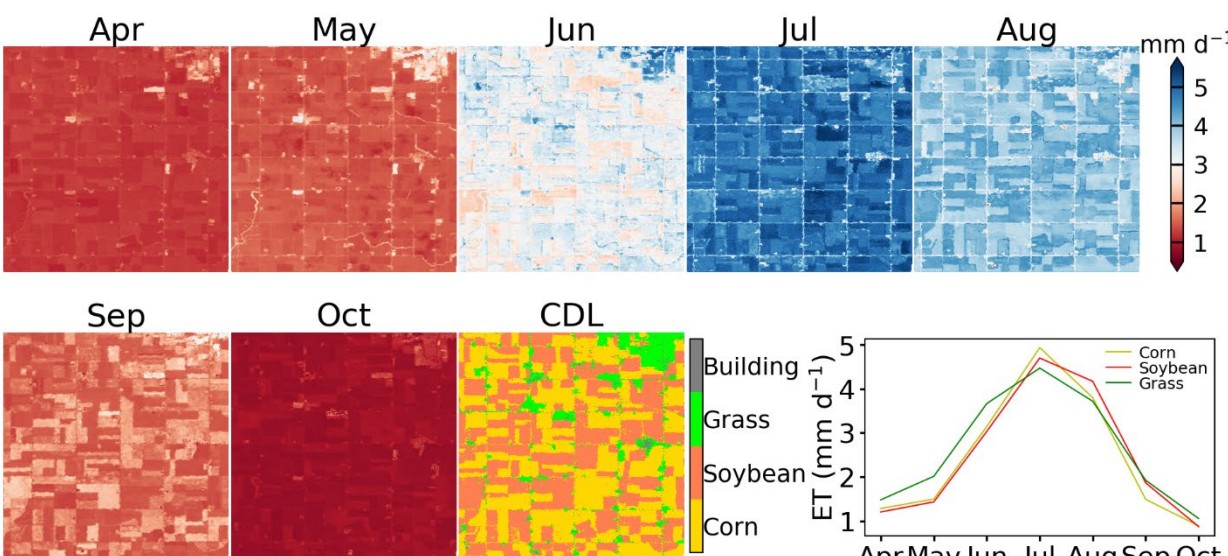

Figure 9. Monthly mean BESS-STAIR ET at Brooks Field (41.9°N – 42.0°N, 96.65°W – 96.75°W) during the growing season of 2000, along with a CDL land cover map. The last subplot shows the average time series of corn, soybean and grass.


Reasonable seasonal cycles for different land cover types are revealed by BESS-STAIR monthly ET averaged from gap-free daily estimations. An example time series of monthly ET maps at Brooks Field during the growing season of 2000 is shown in Figure 9. BESS-STAIR ET clearly captures the temporal dynamics throughout the growing season. All vegetation show low values (e.g., < 2 mm d$^{-1}$) in April, May, September and October, but high values in June, July and August (JJA), with

their peaks in July. Different seasonal cycles for corn, soybean and grass are also captured. Grass has the highest ET among the three vegetation types from April through June. Corn has higher ET than soybean in June and July, and decreases quickly since August. Soybean has the lowest ET from April through June, but has the highest ET in August.




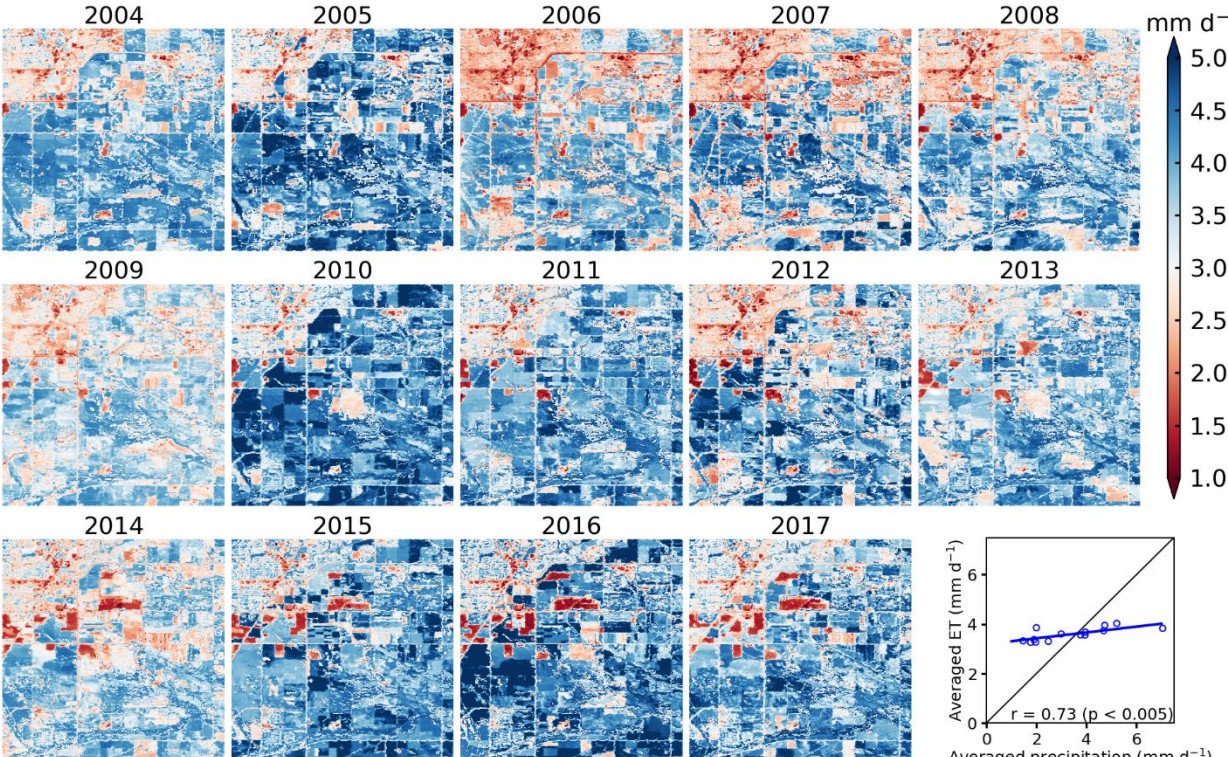

Figure 10. Monthly mean BESS-STAIR ET in July at Rosemount (44.65°N – 44.75°N, 93.05°W – 93.15°W) from 2004 throughout 2017, along with a scatter plot between regional-averaged monthly mean precipitation and ET in July over the 14 years. Monthly precipitation maps are from PRISM (http://www.prism.oregonstate.edu/historical/).

BESS-STAIR is also able to produce long-term ET estimations due to its high computational efficiency. Figure 10 shows an example time series of monthly ET in July at Rosemount from 2004 throughout 2017. The interannual variability is observable, although much smaller than seasonal variability (Figure 10). In this case, time-series of BESS-STAIR ET is in line with that of precipitation, as indicated by a significant linear correlation r = 0.73 (p < 0.005). It is also noted that ET is relatively steady given the high interannual variation in precipitation, and BESS-STAIR enables the investigation of such response at field scales.

## 4 Discussions

### 4.1 Performance of BESS-STAIR ET

In this study, we have presented BESS-STAIR, a new framework for estimating croplands ET at field and daily scale, and we have demonstrated its high performance in the U.S. Corn Belt. The process-based biophysical model BESS, driven by 30-m resolution vegetation-related variables derived from STAIR fused surface spectral reflectance data (Figure 3) and medium resolution environmental inputs derived from MODIS and other satellite data (Figure 1), is able to produce gap-free





ET and PET estimations on field-scale and at daily interval across space and time (Figure 4-7). Over the 12 sites across the U.S. Corn Belt (Figure 2), BESS-STAIR explains 75% variations in flux tower measured daily LE (Figure 4), with an overall RMSE of 2.29 MJ m$^{-2}$ d$^{-1}$ (equivalent to 0.93 mm d$^{-1}$ or 26 W m$^{-2}$), a 27.9% relative error, and stable performance
across sties (Figure 5), as well as consistent seasonal dynamics with respect to flux tower measurements (Figure 6-7).

The error statistics of BESS-STAIR are commensurate with previous high resolution croplands ET mapping studies. Typical RMSE values include 25 W m$^{-2}$ by TSEB-DTD (Guzinski et al., 2014), 35 W m$^{-2}$ by METRIC (Irmak et al., 2011), 62 W m$^{-2}$ by SEBS (McCabe and Wood, 2006), 0.60 mm d$^{-1}$ by SSEBop (Senay et al., 2016), and 1.04 mm d$^{-1}$ by SEBAL (Singh et
al., 2008). Nevertheless, it is worth mentioning that those studies used original Landsat data and therefore suffered from considerably large data gaps. In contrast, BESS-STAIR uses daily Landsat-MODIS fusion data free from any gaps, which leads to temporally continuous ET estimation at the field level, thus can meet the requirements of precision agriculture. In addition, it is worth mentioning that BESS-STAIR is calibration-free and therefore is scalable. It also indicates that the accuracy of BESS-STAIR ET is likely to further improve by using locally optimized driving force or parameter values.

BESS-STAIR is also comparable to other croplands ET mapping studies utilizing data fusion techniques. For example, DisALEXI-STARFM daily ET estimates were validated against the flux tower measurements over the three Mead sites (Yang et al., 2018). They reported error statistics around 1.2 mm d$^{-1}$ RMSE and 29% relative error. BESS-STAIR's performance at these three same sites shows an average of 0.89 mm d$^{-1}$ RMSE and 25.3% relative error (Figure 11). At
monthly scale, the average RMSE and relative errors are only 0.48 mm d$^{-1}$ and 14.3% (Figure A1). In addition, BESS-STAIR has a potential to apply to any croplands around the world back to 1984 when both high spatial resolution data (e.g., Landsat/TM) and high temporal resolution data (e.g., NOAA/AVHRR) were available.

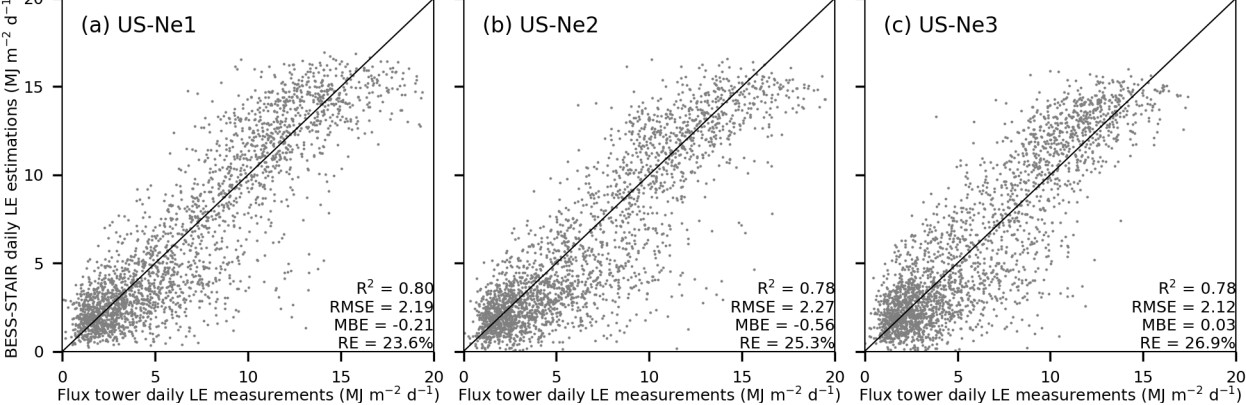

Figure 11. Scatter plots between LE measurements and LE estimations at three sites US-Ne1, US-Ne2 and US-Ne3.





### 4.2 Scientific advantages of BESS-STAIR ET

The efficacy of BESS-STAIR lies in several aspects. First, BESS is a water-carbon-energy coupled biophysical model.
BESS employs atmospheric and canopy radiative transfer modules, carbon assimilation module, stomatal conductance module, and energy balance module (Jiang and Ryu, 2016; Ryu et al., 2011). BESS integrates the simulation of carbon cycle, water cycle and energy cycle in the same framework. Such carbon-water-energy coupling strategy realistically and coherently simulates plant physiology and their response to the environment, specifically the carbon uptake and water loss by plants have been simulated synchronously through environmental constraints on stomatal conductance, with further
constraints by available energy (Baldocchi and Meyers, 1998; Leuning et al., 1995). Many land surface models have already adopted such strategy and have successfully simulated the evolution of terrestrial ecosystems (Ju et al., 2006; Sellers, 1997; Tian et al., 2010). However, this is not the case in commonly-used remote sensing models. Empirical methods, water balance methods, and Priestley–Taylor methods only focus on the water cycle. Energy balance methods, triangular space methods, and Penman–Monteith methods couple water cycle and energy cycle and consider ET in the context of energy partitioning.
BESS, unlike these remote sensing models, constrains ET with regards to both energy requirement and carbon requirement, thanks to explicit modeling of radiative transfer and stomatal behavior processes. For above reasons, BESS-STAIR ET does not only achieve high accuracy (Figure A1 – A3), but also accurately capture responses to GPP, radiation, temperature, and humidity at daily scale (Table 3). Thus, BESS-STAIR has the potential to advance the understanding of crop responses to climate change through bridging remote sensing data and land surface models, which was first suggested by Sellers et al.
(1997) more than 20 years ago.

Table 3. BESS-STAIR captures the correct response of daily LE to GPP, radiation (Rg), temperature (Ta) and humidity (VPD) as compared to flux tower measurements over the three Mead sites from 2001 through 2012. The linear equation slopes and correlation coefficients between LE and other factors are similar in flux tower measurements and BESS-STAIR
estimations, for both the whole growing season (April – October) or only peak growing season (June, July and August). For "flux tower" columns, Rg, Ta and VPD are from site measurements, while for "BESS-STAIR" columns, they are from satellite-derived coarse resolution inputs.

| Time period | Relationship | Flux tower | | BESS-STAIR | |
|---|---|---|---|---|---|
| | | Equation | Correlation | Equation | Correlation |
| Growing season (April – October) | LE (MJ m$^{-2}$ d$^{-1}$) ~ GPP (gC m$^{-2}$ d$^{-1}$) | y = 0.44x + 3.19 | r = 0.86 (p < 0.01) | y = 0.65x + 2.60 | r = 0.90 (p < 0.01) |
| | LE (MJ m$^{-2}$ d$^{-1}$) ~ Rg (MJ m$^{-2}$ d$^{-1}$) | y = 0.33x - 0.07 | r = 0.56 (p < 0.01) | y = 0.29x + 0.28 | r = 0.51 (p < 0.01) |
| | LE (MJ m$^{-2}$ d$^{-1}$) ~ Ta (°C) | y = 0.44x - 1.60 | r = 0.67 (p < 0.01) | y = 0.43x - 1.64 | r = 0.54 (p < 0.01) |
| | LE (MJ m$^{-2}$ d$^{-1}$) ~ VPD (hPa) | y = 0.32x + 4.10 | r = 0.31 (p < 0.01) | y = 0.44x - 2.50 | r = 0.68 (p < 0.01) |
| JJA (June, July, August) | LE (MJ m$^{-2}$ d$^{-1}$) ~ GPP (gC m$^{-2}$ d$^{-1}$) | y = 0.35x + 4.85 | r = 0.76 (p < 0.01) | y = 0.39x + 6.36 | r = 0.63 (p < 0.01) |
| | LE (MJ m$^{-2}$ d$^{-1}$) ~ Rg (MJ m$^{-2}$ d$^{-1}$) | y = 0.36x + 1.79 | r = 0.58 (p < 0.01) | y = 0.36x + 3.40 | r = 0.77 (p < 0.01) |
| | LE (MJ m$^{-2}$ d$^{-1}$) ~ Ta (°C) | y = 0.52x - 2.20 | r = 0.46 (p < 0.01) | y = 0.15x + 9.40 | r = 0.28 (p < 0.01) |





| LE (MJ m$^{-2}$ d$^{-1}$) ~ VPD (hPa) | y = 0.28x + 7.61 | r = 0.32 (p < 0.001) | y = 0.28x + 4.37 | r = 0.30 (p < 0.001) |
| --- | --- | --- | --- | --- |

The second strength is that BESS-STAIR is designed most sensitive to the variables which can be well-quantified from remote sensing data. BESS ET is most sensitive to solar radiation, followed by LAI (Ryu et al., 2011), as BESS ET is mainly constrained by net radiation and GPP. In most cases, solar radiation is the predominant component of net radiation, while LAI determines the capacity of radiation absorption and subsequently determines GPP. BESS explicitly computes radiation components in high accuracy by driving an atmosphere radiative transfer model FLiES using MODIS cloud, aerosol and atmospheric profile products. Globally, BESS-estimated solar radiation has its R$^2$ about 0.85 and 0.95 for MODIS snapshots

and 4-day averages, respectively (Ryu et al., 2018). On the other hand, BESS-STAIR calculates high spatiotemporal resolution LAI and albedo from fused surface reflectance data. Since Landsat and MODIS surface reflectance products are publicly-available and highly-reliable (Claverie et al., 2015; Masek et al., 2006), spatial heterogeneity and temporal dynamics of crop growing conditions are well captured (Figure 8). This study only uses reflectance data fused from Landsat and MODIS, but STAIR can be easily extended to further incorporate other types of data, such as Sentinel-2 (10 m

resolution) and Planet Lab CubeSats (3 m resolution) (McCabe et al., 2017). By incorporating more high resolution observations, the relevance of reconstructed high resolution image series can be further improved.

The third strength is that BESS-STAIR is able to perform under all-weather conditions. BESS-STAIR fills data gaps in surface reflectance, which has a smooth day-to-day variation even with changes in sky conditions (Liu et al., 2017). Based

on filtered surface reflectance, LAI and albedo time series are well-reconstructed, and subsequently BESS-STAIR could directly work under all-weather condition. In this manner, BESS-STAIR has no need to fill cloudy-sky ET using clear-sky ET estimations, which is error prone because the empirically-filled ET estimations usually lack sophisticated process-level model constraints and thus can have large uncertainties. Figure 12 shows that the estimation errors of BESS-STAIR ET do not change significantly under different sky conditions, with low to high "sky clearness index" referring more cloudy to

more clear sky conditions.





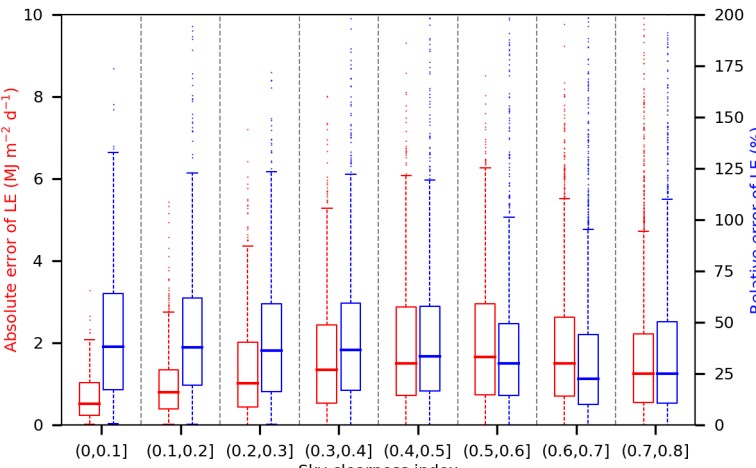

Figure 12. BESS-STAIR estimated daily LE has similar performance with varying sky clearness index (the ratio of incoming radiation on surface to that on top-of-atmosphere). The lower and upper boundaries of boxes refer to the first and third quartile of error statistics. The bars inside boxes refer to median values. The whiskers indicate 1.5 times of distance between the first and third quartiles.

### 4.3 Limitations and future improvements of BESS-STAIR ET

In this study, several inputs used by BESS have some limitations in terms of generality and accessibility. First, three plant functional parameters, peak $V_{cmax25}$, Ball-Barry slope and intercept are obtained from literatures, assuming constant given C3 or C4 plant type. Other land surface models tend to use the similar strategy by assigning fixed values to a given plant functional type (PFT) (Bonan et al., 2011; Kattge et al., 2009; Miner et al., 2017). The drawback of this strategy is the overlook of within-PFT variations and the feedback mechanisms between vegetation and its environment (Van Bodegom et al., 2014). These limitations might be mitigated by incorporating innovative leaf trait estimation techniques emerged in recent years, such as imaging spectroscopy (Serbin et al., 2015), sun-induced fluorescence (Zhang et al., 2018), and plant optimization theory (Walker et al., 2017; Wang et al., 2017). Second, BESS-STAIR in this study uses CDL data which is only available in United States. Fortunately, BESS does not require specific crop types but only C3/C4 distributions, and the separation of the major C4 crop maize from other crops is practical using time-series satellite data (Cai et al., 2018; Zhong et al., 2016). It is noted that misclassification of C4 and C4 crops are likely to cause large bias in GPP, but relatively small bias in ET (Fig. A1 – A3).

Though BESS-STAIR is able to capture water stress impact on ET in the U.S. Corn Belt where atmospheric demands play a major role, its applicability to regions where soil supply dominates needs further investigation. Some studies suggest that optical signal as an indicator of drought performs at a longer time scale than thermal signal does (Otkin et al., 2017). Drought first decreases soil moisture content due to enhanced ET induced by high atmospheric demand, then decreases ET



due to low soil moisture content, and finally causes damage to plants which changes surface reflectance. Accordingly, LAI may not serve as a relevant early warning of droughts. Furthermore, severe soil moisture stress may cause physiological deterioration in addition to structural damage that has been reflected in LAI. To address this issues for dry regions, we acknowledge that LST observations may provide essential adding values. At this point, the capacity of BESS-STAIR in estimating LST leads to a possibility of optimizing BESS-STAIR using satellite-derived LST. Recent advances of innovative thermal observation platforms such as ECOSTRESS (Hulley et al., 2017), GOES-R (Schmit et al., 2017), and Sentinel-3 (Zheng et al., 2019) have provide great opportunity to integrate satellite-derived LST with the BESS-STAIR.

The BESS model itself in essence estimates instantaneous ET. The ratio of snapshot potential solar radiation to daily potential solar radiation is adopted as a scaling factor for the temporal upscaling of ET (Ryu et al., 2012). In this study, BESS runs two times per day, utilizing radiation components derived from Terra/MODIS (around 11:00 AM) and Aqua/MODIS (around 1:00 PM) data, respectively. The two instantaneous ET estimates are separately upscaled to daily estimates and averaged. In spite of robustness of the upscaling algorithm (Ryu et al., 2012), bias cannot be avoided if the sky conditions at two overpass times are not representative for that day, which is natural and common in the presence of moving cloud. Since BESS is a time-independent model and can perform at any time during daytime, adding more snapshots to account for the diurnal variations of radiation can solve this problem. Unfortunately, fine-resolution polar-orbiting satellite usually have similar overpass times (10:00 AM – 11:00 AM and 1:00 PM – 2:00 PM), so even adding more satellites is likely to bring redundant information only. Reanalysis radiation data covering diurnal cycle have limited accuracy and coarse resolution (Babst et al., 2008; Zhang et al., 2016b), so they may be unable to provide much added values as well. Next-generation geostationary satellites, acquiring data with both high spatial and high temporal resolutions such as GOES-R and GaoFen-4 (Goodman et al., 2012; Xu et al., 2017), are expected to enable BESS-STAIR ET in hourly or sub-hourly interval and subsequently generate more realistic daily ET estimates.

## 5 Conclusions

In this study we presented BESS-STAIR, a new framework to estimate high spatiotemporal resolution ET that can be used for field-level precision water resources management. BESS-STAIR couples a satellite-driven water-energy-carbon coupled biophysical model BESS with a generic and fully-automated fusion algorithm STAIR to generate gap-free 30-m resolution daily ET estimations. Comprehensive evaluation of BESS-STAIR ET estimations revealed: 1) reliable performance over 12 flux tower sites across the U.S. Corn Belt, and 2) reasonable spatial patterns, seasonal cycles and interannual dynamics. The proposed BESS-STAIR framework has demonstrated its ability to provide significant advancements with regard to daily field-level estimations of ET at regional and decadal scales. We expect BESS-STAIR to become a solid tool for precision water resources management and other precision agriculture applications for the U.S. Corn Belt as well as other agricultural areas around the world, thanks to the global coverage of input data.



Data availability. The data generated in this study are available upon request.

Competing interests. The authors declare that they have no conflict of interest.

Author contribution. C.J. and K.G. designed the study, C.J. conducted the modeling and analysis, M.P. and B.P. provided
research inputs during the analysis, Y.R. and S.W. provided guidance in the BESS model and the STAIR algorithm,
respectively. All the authors contributed to the writing of the manuscript.

**Acknowledgements**

C.J. and K.G. were funded by the DOE Center for Advanced Bioenergy and Bioproducts Innovation (U.S. Department of
Energy, Office of Science, Office of Biological and Environmental Research under Award Number DE-SC0018420). Any
opinions, findings, and conclusions or recommendations expressed in this publication are those of the author(s) and do not
necessarily reflect the views of the U.S. Department of Energy. K.G., B. P., and S.W. are funded by NASA awards
(NNX16AI56G and 80NSSC18K0170) and USDA National Institute of Food and Agriculture (NIFA) Foundational Program
award (2017-67013-26253, 2017-68002-26789, 2017-67003-28703). The development of the BESS model was mainly
supported by The National Research Foundation of Korea (NRF-2014R1A2A1A11051134). K.G. and C.J. also acknowledge
the support from Blue Waters Professorship from National Center for Supercomputing Applications of UIUC. This research
is part of the Blue Waters sustained-petascale computing project, which is supported by the National Science Foundation
(awards OCI-0725070 and ACI-1238993) and the state of Illinois. Blue Waters is a joint effort of the University of Illinois at
Urbana-Champaign and its National Center for Supercomputing Applications. We thank the U.S. Landsat project
management and staff at USGS Earth Resources Observation and Science (EROS) Center South Dakota for providing the
Landsat data free of charge. We also thank NASA freely share the MODIS products.





**Appendix**

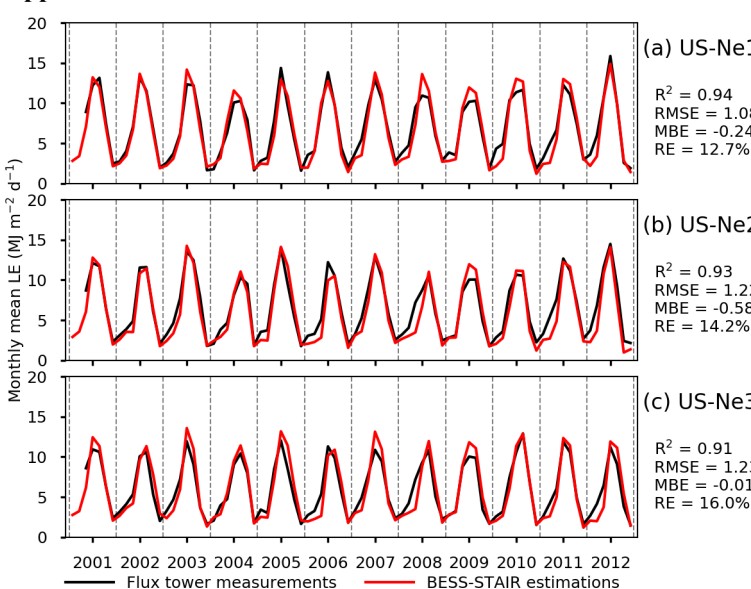

Figure A1. Time series of monthly mean LE from flux tower measurements and BESS-STAIR estimations.

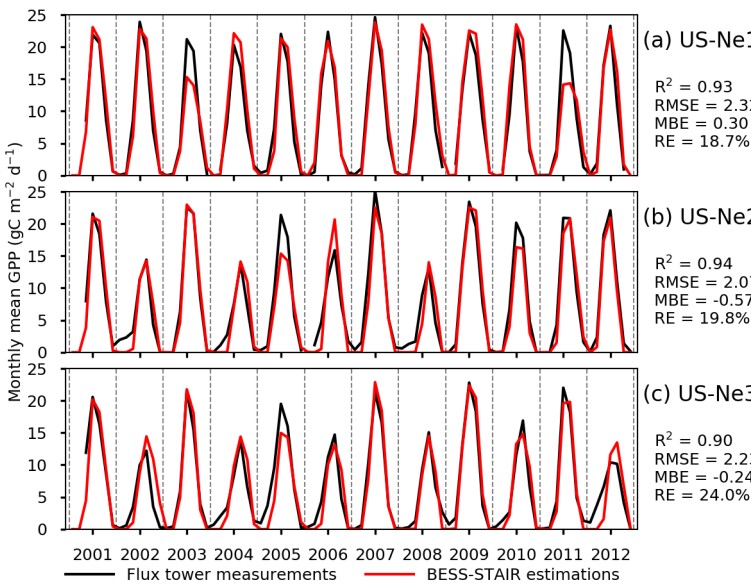

Figure A2. Time series of monthly mean GPP from flux tower measurements and BESS-STAIR estimations. Significant underestimations in 2003 and 2011 for Ne1, in 2005 for Ne2, and in 2005 for Ne3 are due to misclassification of corn as soybean in CDL. Significant overestimations in 2006 for Ne2 are due to misclassification of soybean as corn in CDL.


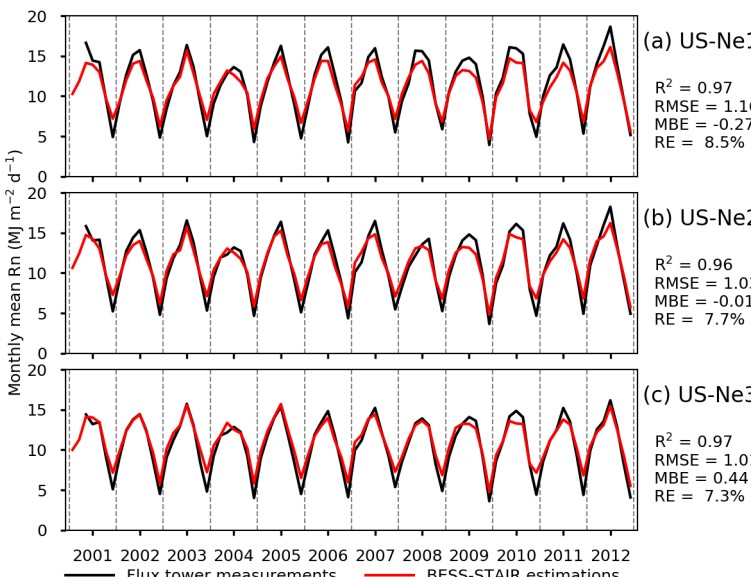

Figure A3. Time series of monthly mean Rn from flux tower measurements and BESS-STAIR estimations.

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
