# Peer review of "BESS-STAIR: a framework to estimate daily, 30-meter, and all-weather crop evapotranspiration using multi-source satellite data for the U.S. Corn Belt"

_Hydrology and Earth System Sciences, 2019_

## Referee Comment (RC1) · Anonymous Referee #1 · 2 Sep 2019

General comments: The study "BESS-STAIR: a framework to estimate daily, 30-meter, and all-weather crop evapotranspiration using multi-source satellite data for the U.S. Corn Belt" used the BESS model with 30-m fused inputs from a automated data fusion algorithm (STAIR) to simulate fine resolution cropland ET across the U.S. Corn Belt from 2000 to 2017. The results showed good performance compared with field measurement, indicating that BESS-STAIR is applicable for field scale ET simulations in the U.S. Corn Belt, which is useful and meaningful for agricultural water management and precision agriculture applications. The manuscript was well written and followed

a good logic. Lots of work was done by the authors to explore applicable and reliable methods for agricultural management. It is good for the farmers and decision makers to know agricultural water demands. Thus, I think this study could be considered for publication in this journal. However, I have several minor comments listed below.

Specific comments: 1. Please use only one term (ET or LE) consistently throughout the manuscript. It would be easier for people to read this manuscript. 2. In the section 2.1, please list the most important equations in the BESS model for ET calculations. And please list the full name of the variables before using the abbreviations (e.g. $\alpha$VIS, Vcmax, etc.) 3. Ln. 205, please show the equation for clear demonstration. 4. Ln. 254, CI could not be found in Eq. 6 or 7. Please check the manuscript clearly to correct this kind of errors. 5. Fig. 6, is the irrigation measured in the field? In Fig. 8 and 9, it would be good to point out flux tower sites. 6. I think Fig. 7 could be put in the supplementary materials, since Fig. 6 already showed good performance of BESS-STAIR ET. 7. Ln. 346-347, the authors said "measured daily LE do not show strong and fast response to precipitation and/or irrigation", however, in Fig. 10, averaged ET showed significant correlations with precipitation. How did the author interpret this?

---

## Referee Comment (RC2) · Anonymous Referee #2 · 6 Nov 2019

General: This is an interesting and impactful study. However, I think that the authors have not currently highlighted its strengths as well as they could. The way I see it, this study provides a novel ET algorithm because of (1) the explicit treatment and modeling of stomatal conductance and canopy conductance; (2) the detailed representation of leaf architecture and inclination distribution; and (3) the ingenuity in gap filling cloudy days in the Midwest. I think it is disingenuous of the authors to call this a coupled water-carbon-energy model. To me, this implies that there will be some sort of simulation of photosynthesis, NEP, NPP, and/or yields. I see that stomatal conductance

was calculated using the Farquhar and Ball-Berry models. In order to make these calculations, there needed to be an estimate of net photosynthesis. Why are these estimates and maps not included as results in the study? If there is some limitation to simulating the carbon cycle, the authors should explain what that is or actually simulate some carbon components. I'm guessing there is some limitation here and I think that rather than calling BESS-STAIR a water-carbon-energy model, that the authors should focus on the excellent strides that they have made by explicitly simulating the dynamic stomatal response, leaf inclination, and gap filling using surface albedo. Specific comments: Line 21: "water-carbon-energy" coupled is a little disingenuous. To me this implies there will be a carbon budget or some carbon-related outputs from the model (e.g. Anet, NEP, NPP, yield). Line 22: 'satellite' spelled incorrectly. Line 37: Add "evaporation" to leaf transpiration. Line 37: "ET at cropland is usually considered as crop water needs" is confusing. Please rephrase. Line 43: USDA, 1997 is an outdated reference. Please update, perhaps with some Midwest US specific references about precision irrigation. Line 44: You bring up the urgency of this need here. I ask you to follow up on this both in your computation and the discussion of the scalability of this model and also how it compares or outcompetes other similar models (e.g. STAR-FM). Line 48: "competitions" is an odd work to use here. Please rephrase. Line 113: Figure 1 is where I get confused about your use of coupled water-carbon-energy cycling to describe this model. If you have VCmax 25 and Ball-Berry parameters, what are you missing for Amax, and other carbon outputs? Line 155: How is STAIR different from STAR-FM? (Semmens et al., 2015) Lines 197-199: This is the first time that corn and soy are mentioned as the specific crops studied. It would be better if these were specified earlier in the manuscript. Lines 235-235: Including the leaf inclination distribution is novel and interesting. It would be great to highlight this aspect of your work more in the abstract and also place it into context in the intro (I do not believe other satellite-based ET models characterize the canopy to this detail). Line 255: Is soil albedo also estimated prior to canopy closure? Line 322: I agree with Reviewer 1. Please make the necessary conversions and stick with ET. Introducing LH at this point in the paper
is a distraction. Line 394: Please expand on the "high computational efficiency" of this model. Line 407: You mention PET here, but there are no PET maps. It would be interesting to compare PET to ET in a drought year (e.g. 2012). Line 443: You criticize other models for only focusing on the water cycle, but as it stands, this paper also only shares results related to the water cycle. I think you either should share some carbon cycle results or remove this type of language. Lines 478-479: Traditionally, we do not introduce new figures in the discussion section. It would be better to move Figure 12 to the results section. Line 505: Can you investigate drought and soil moisture using the 2012 drought year in your dataset? Line 531-534: Back to your urgency point in the introduction—how far are we from real-time ET estimates at 30-m being freely available for irrigation management?

---

## Author Comment (AC1) · 4 Dec 2019

1. The study "BESS-STAIR: a framework to estimate daily, 30-meter, and all-weather crop evapotranspiration using multi-source satellite data for the U.S. Corn Belt" used the BESS model with 30-m fused inputs from a automated data fusion algorithm (STAIR) to simulate fine resolution cropland ET across the U.S. Corn Belt from 2000 to 2017. The results showed good performance compared with field measurement, indicating that BESS-STAIR is applicable for field scale ET simulations in the U.S. Corn Belt, which is useful and meaningful for agricultural water management and precision

agriculture applications. The manuscript was well written and followed a good logic. Lots of work was done by the authors to explore applicable and reliable methods for agricultural management. It is good for the farmers and decision makers to know agricultural water demands. Thus, I think this study could be considered for publication in this journal.

–> Dear reviewer: we greatly appreciate your positive summary on our manuscript.

2. Please use only one term (ET or LE) consistently throughout the manuscript. It would be easier for people to read this manuscript.

–> We have accepted your suggestion and made changes throughout text, figures and tables.

3. In the section 2.1, please list the most important equations in the BESS model for ET calculations. And please list the full name of the variables before using the abbreviations (e.g. aVIS, Vcmax, etc.)

–> We appreciate your suggestion. The most important equations in the BESS model for ET calculations are now listed in Appendix 1 in the revised version (L570). Full names of LAI, $\alpha$VIS and $\alpha$NIR were listed in L90. Full name of Vcmax25 was shown in L120.

4. Ln. 205, please show the equation for clear demonstration.

–> We have followed your suggestion. We have added a Table 1 "Linear equations for LAI (y) as a function of VI (x) for corn, soybean, and the combination of corn and soybean".

[Table 1]

5. Ln. 254, CI could not be found in Eq. 6 or 7. Please check the manuscript clearly to correct this kind of errors.

–> Thanks for the suggestion. We have deleted this sentence. CI is not used.

6. Fig. 6, is the irrigation measured in the field?

–> Yes. The irrigation data at US-Ne1 and US-Ne2 are provided by FLUXNET2015 database.

7. In Fig. 8 and 9, it would be good to point out flux tower sites.

–> We have followed your suggestion by adding flux tower marks in all maps.

8. I think Fig. 7 could be put in the supplementary materials, since Fig. 6 already showed good performance of BESS-STAIR ET.

–> We agree that key information have already been shown in Fig. 6 so we have removed this figure.

9. Ln.346-347, the authors said "measured daily LE do not show strong and fast response to precipitation and/or irrigation", however, in Fig. 10, averaged ET showed significant correlations with precipitation. How did the author interpret this?

–> We appreciate this important point. In most areas in the U.S. Corn Belt, atmospheric water demand dominates ET over soil water supply. In fact, the original Fig. 10 was special because that area-month has the highest correlation between ET and precipitation, thus that figure was not representative and misleading. We have replaced it by another figure:

[Figure 10]

Figure 10. Peak growing season (June, July and August) BESS-STAIR ET/PET at Bondville (39.95°N – 40.05°N, 88.25°W – 88.35°W) from 2001 throughout 2017, along with two scatter plots between peak growing season precipitation and ET/PET and peak growing season VPD and ET/PET over the 17 years. Precipitation and VPD data are from Daily Surface Weather Data (Daymet) at Illinois Bondville, where VPD is derived using maximum air temperature and water vapor pressure. Circles indicate flux towers in this region. Accordingly, we have rewritten the paragraph (L398): "BESS-

STAIR is also able to produce long-term ET/PET estimation as an indicator of drought. Figure 10 shows an example time series of peak growing season ET/PET at Bondville from 2001 through 2017. Overall substantial interannual variability is shown, with regional average ET/PET values ranging from the 0.76 in an extremely dry year 2012 to 0.91 in an extremely wet year 2015. A positive linear relationship ($r = 0.42$, $p < 0.1$) is observed between BESS-STAIR ET/PET and precipitation, and a negative linear relationship ($r = -0.58$, $p < 0.05$) is observed between BESS-STAIR ET/PET and VPD. The relative stronger relationship between ET/PET and VPD than that between ET/PET and precipitation indicates atmospheric water demand is likely to contribute more to drought than soil water supply in this area."
* * *
[Figure]

**Fig. 1.** Figure 10

| VI | Corn | Soybean | Combination |
|---|---|---|---|
| WDRVI | y = 6.288x + 4.631 | y = 4.584x + 3.432 | y = 5.745x + 4.288 |
| GWDRVI | y = 8.964x + 5.875 | y = 6.384x + 4.275 | y = 8.110x + 5.395 |
| EVI | y = 10.569x - 2.165 | y = 8.116x - 1.936 | y = 9.665x - 1.993 |
| LSWI | y = 9.156x + 1.070 | y = 7.553x + 0.888 | y = 8.944x + 0.982 |

**Fig. 2.** Table 1

---

## Author Comment (AC2) · 4 Dec 2019

1. The way I see it, this study provides a novel ET algorithm because of (1) the explicit treatment and modeling of stomatal conductance and canopy conductance; (2) the detailed representation of leaf architecture and inclination distribution; and (3) the ingenuity in gap filling cloudy days in the Midwest.

–> Dear reviewer: we greatly appreciate your positive comments on our study.

2. I think it is disingenuous of the authors to call this a coupled water-carbon-energy

model. To me, this implies that there will be some sort of simulation of photosynthesis, NEP, NPP, and/or yields. I see that stomatal conductance was calculated using the Farquhar and Ball-Berry models. In order to make these calculations, there needed to be an estimate of net photosynthesis. Why are these estimates and maps not included as results in the study? If there is some limitation to simulating the carbon cycle, the authors should explain what that is or actually simulate some carbon components. I'm guessing there is some limitation here and I think that rather than calling BESS-STAIR a water-carbon-energy model, that the authors should focus on the excellent strides that they have made by explicitly simulating the dynamic stomatal response, leaf inclination, and gap filling using surface albedo.

–> We are glad that you see the water-carbon-energy integrated model as an innovation in high resolution ET estimation. Although the model estimates ET, GPP and Rnet simultaneously, the objective of this study is to demonstrate the performance of ET for water resource management. As a result, we purposely avoid putting too many GPP estimation results in the manuscript which might distract readers. Besides, GPP data is unavailable for many validation sites used in this study, and processing GPP measurements would take too much extra space, which would make the current manuscript too long. Even though, in the original manuscript, we still demonstrated good GPP estimation performance at three sites in Figure A1. We also showed that BESS-STAIR GPP∼ET relationship is consistent with flux tower GPP∼ET relationship at three sites in Table 3. Furthermore, readers can refer to our previous paper about BESS global GPP and ET products (Jiang and Ryu 2016). In the revised version, we have added several GPP maps in Figure A4 per your request.

Jiang, C., & Ryu, Y. (2016). Multi-scale evaluation of global gross primary productivity and evapotranspiration products derived from Breathing Earth System Simulator (BESS). Remote Sensing of Environment, 186, 528–547. https://doi.org/10.1016/j.rse.2016.08.030

3. Line 21: "water-carbon-energy" coupled is a little disingenuous. To me this implies

there will be a carbon budget or some carbon-related outputs from the model (e.g. Anet, NEP, NPP, yield).

–> Please refer to the above response.

4. Line 22: 'satellite' spelled incorrectly. –> We have corrected it.

5. Line 37: Add "evaporation" to leaf transpiration. –> We have added it.

6. Line 37: "ET at cropland is usually considered as crop water needs" is confusing. Please rephrase. –> We have rephrased it: "ET at cropland is usually considered as crop water use".

7. Line 43: USDA, 1997 is an outdated reference. Please update, perhaps with some Midwest US specific references about precision irrigation. –> Thanks for the suggestion. We have replaced the reference by a recently published report by The U.S. Government Accountability Office.

8. Line 44: You bring up the urgency of this need here. I ask you to follow up on this both in your computation and the discussion of the scalability of this model and also how it compares or outcompetes other similar models (e.g. STAR-FM).

–> We appreciate this important suggestion. However, the purpose of this manuscript is to propose a new framework and to demonstrate its performance. We only would like to let the community know that this framework differs from existing ones such as DisALEXI-STARFM with regards to methodology. We do not want to comment on or judge others' framework without careful investigation, especially considering their data are not publicly available. Meanwhile, we are open to share our data and welcome a third-party to conduct potential comparison studies and comprehensively evaluate the difference between frameworks in terms of performance, computation, scalability, etc.

9. Line 48: "competitions" is an odd work to use here. Please rephrase. –> We have changed it to "balance".

10. Line 113: Figure 1 is where I get confused about your use of coupled water-carbon-energy cycling to describe this model. If you have VCmax 25 and Ball-Berry parameters, what are you missing for Amax, and other carbon outputs?

–> Thank you for your interests on the carbon part of our framework. Carbon output is not missed. Please refer to our response 2.

11. Line 155: How is STAIR different from STAR-FM? (Semmens et al., 2015)

–> When a single pixel suffers from gaps in two input snapshots of Landsat images, STARFM continues searching for the closest possible Landsat-MODIS image pair of a matching date, even if it is months away from the target prediction date. By comparison, STAIR utilizes the whole time series to conduct fusion. STARFM requires users to manually select and specify clear reference images for the methods. It is possibly such semi-automatic feature leads to the fact that DisALEXI-STARFM has not been investigated at regional scale over long-term span. By comparison, STAIR is completely automatic and thus more scalable. Detailed difference can be found in (Luo et al., 2018).

Luo, Y., Guan, K., & Peng, J. (2018). STAIR: A generic and fully-automated method to fuse multiple sources of optical satellite data to generate a high-resolution, daily and cloud-/gap-free surface reflectance product. Remote Sensing of Environment, 214(March), 87–99. https://doi.org/10.1016/j.rse.2018.04.042

12. Lines 197-199: This is the first time that corn and soy are mentioned as the specific crops studied. It would be better if these were specified earlier in the manuscript.

–> We have followed your suggestion. We have added one sentence in the first paragraph of the manuscript: "In the U.S. Corn Belt where more than 85% of corn and soybean are produced in the U.S. (Grassini et al., 2015) . . ."

13. Lines 235-235: Including the leaf inclination distribution is novel and interesting. It would be great to highlight this aspect of your work more in the abstract and also place

it into context in the intro (I do not believe other satellite-based ET models characterize the canopy to this detail).

–> Thank you for your positive comments. However, we set three fixed types of leaf inclination distribution "spherical", "planophile" and "plagiophile" for corn, soybean, and other biomes, and currently they are only used in the LAI estimation procedure. We think this contribution is not large enough to highlight.

14. Line 255: Is soil albedo also estimated prior to canopy closure?

–> Yes, we used "spectral reflectance in April when no crop is planted across the study area to derive representative soil spectral reflectance" (L238) and estimated visible and near-infrared albedo using Eq. (6) and (7) (L257).

15. Line 322: I agree with Reviewer 1. Please make the necessary conversions and stick with ET. Introducing LH at this point in the paper is a distraction.

–> We have accepted your suggestion.

16. Line 394: Please expand on the "high computational efficiency" of this model.

–> This paragraph demonstrates interannual variations of BESS-STAIR ET, and computational efficiency is not key message we would like to convay here. Therefore, we have removed this statement to avoid distraction.

17. Line 407: You mention PET here, but there are no PET maps. It would be interesting to compare PET to ET in a drought year (e.g. 2012).

–> We appreciate and have followed your suggestion. In the revised manuscript, we have added two new figures about PET and corresponding paragraphs:

[Figure 7]

Figure 7. Seasonal time series of daily ET/PET derived from BESS-STAIR and flux tower for US-Ne1, US-Ne2, and NS-Ne3 in 2012, along with measured daily mean soil

water content (SWC).

"Figure 7 shows the comparison between BESS-STAIR ET/PET and flux tower ET/PET at three sites (US-Ne1, US-Ne2, and NS-Ne3) at Mead, Nebraska. Overall, BESS-STAIR agrees well with flux tower in both magnitude and seasonal cycle. Although 2012 is a severe drought year, soil water content (SWC) at the rainfed site US-Ne3 still shows a relatively high level (> 0.2). As a result, ET/PET from both BESS-STAIR and flux tower are at the same level with the adjacent two irrigated sites (US-Ne2 and US-Ne3)."

[Figure 10]

Figure 10. Peak growing season (June, July and August) BESS-STAIR ET/PET at Bondville (39.95°N – 40.05°N, 88.25°W – 88.35°W) from 2001 throughout 2017, along with two scatter plots between peak growing season precipitation and ET/PET and peak growing season VPD and ET/PET over the 17 years. Precipitation and VPD data are from Daily Surface Weather Data (Daymet) at Illinois Bondville, where VPD is derived using maximum air temperature and water vapor pressure. Circles indicate flux towers in this region.

"BESS-STAIR is also able to produce long-term ET/PET estimation as an indicator of drought. Figure 10 shows an example time series of peak growing season ET/PET at Bondville from 2001 through 2017. Overall substantial interannual variability is shown, with regional average ET/PET values ranging from 0.76 in an extremely dry year 2012 to 0.91 in an extremely wet year 2015. A positive linear relationship ($r = 0.42$, $p < 0.1$) is observed between BESS-STAIR ET/PET and precipitation, and a negative linear relationship ($r = -0.58$, $p < 0.05$) is observed between BESS-STAIR ET/PET and VPD. The relative stronger relationship between ET/PET and VPD than that between ET/PET and precipitation indicates atmospheric water demand is likely to contribute more to drought than soil water supply in this area."

18. Line 443: You criticize other models for only focusing on the water cycle, but as it

stands, this paper also only shares results related to the water cycle. I think you either should share some carbon cycle results or remove this type of language.

–> We have shared some carbon cycle results in the manuscript. Please refer to our response 2.

19. Lines 478-479: Traditionally, we do not introduce new figures in the discussion section. It would be better to move Figure 12 to the results section.

–> While we greatly appreciate the reviewer's suggestion, we feel that we have some different opinions on this point. We think that the Discussion section could have a figure, as long as the contents are to support interpreting findings, placing them in a bigger context, and relating them to other work, any measures. Here Figure 12 is to interpret WHY BESS-STAIR ET is advanced, whereas the whole results section is to show WHAT BESS-STAIR ET looks like. Therefore, we think putting it in the discussion section fits well for the overall discussion part. We are also aware that many papers in this journal and other top journals have new figures introduced in the discussion section, e.g.:

Guzinski, R., Nieto, H., Stisen, S., & Fensholt, R. (2015). Inter-comparison of energy balance and hydrological models for land surface energy flux estimation over a whole river catchment. Hydrology and Earth System Sciences, 19(4), 2017–2036. https://doi.org/10.5194/hess-19-2017-2015

Gerken, T., Bromley, G. T., Ruddell, B. L., Williams, S., & Stoy, P. C. (2018). Convective suppression before and during the United States Northern Great Plains flash drought of 2017. Hydrology and Earth System Sciences, 22(8), 4155–4163. https://doi.org/10.5194/hess-22-4155-2018

Konings, A. G., & Gentine, P. (2017). Global variations in ecosystem-scale isohydricity. Global Change Biology, 23(2), 891–905. https://doi.org/10.1111/gcb.13389

Huemmrich, K. F., Campbell, P., Landis, D., & Middleton, E. (2019). Developing a common globally applicable method for optical remote sensing of ecosystem light use efficiency. Remote Sensing of Environment, 230(July 2018), 111190. https://doi.org/10.1016/j.rse.2019.05.009

20. Line 505: Can you investigate drought and soil moisture using the 2012 drought year in your dataset?

–> We have accepted your suggestion. Please refer to our response 17.

21. Line 531-534: Back to your urgency point in the introduction: are we from real-time ET estimates at 30-m being freely available for irrigation management?

–> No, we haven't achieved real-time ET estimations in this study. This is indeed our ultimate goal and still on the development. We are developing in-season classification of corn and soybean which is required by BESS as well as in-season STAIR fusion.

[Figure]

**Fig. 1.** Figure 7

[Figure]

**Fig. 2.** Figure 10